

# Brain hothubs and dark functional networks: correlation analysis between amplitude and connectivity for Broca's aphasia

Feng Lin[1,2], Shao-Qiang Cheng[3], Dong-Qing Qi[4], Yu-Er Jiang[1], Qian-Qian Lyu[1], Li-Juan Zhong[1] and Zhong-Li Jiang[1,2]

[1] Department of Rehabilitation Medicine, The First Affiliated Hospital of Nanjing Medical University, Nanjing, Jiangsu, China
[2] Department of Rehabilitation Medicine, The Affiliated Sir Run Run Hospital of Nanjing Medical University, Nanjing, Jiangsu, China
[3] Department of Neurology, The First People's Hospital of Xianyang, Xianyang, Shananxi, China
[4] The First Affiliated Hospital of USTC, Division of Life Sciences and Medicine, University of Science and Technology of China, Hefei, Anhui, China

## ABSTRACT

Source localization and functional brain network modeling are methods of identifying critical regions during cognitive tasks. The first activity estimates the relative differences of the signal amplitudes in regions of interest (ROI) and the second activity measures the statistical dependence among signal fluctuations. We hypothesized that the source amplitude–functional connectivity relationship decouples or reverses in persons having brain impairments. Five Broca's aphasics with five matched cognitively healthy controls underwent overt picture-naming magnetoencephalography scans. The gamma-band (30–45 Hz) phase-locking values were calculated as connections among the ROIs. We calculated the partial correlation coefficients between the amplitudes and network measures and detected four node types, including hothubs with high amplitude and high connectivity, coldhubs with high connectivity but lower amplitude, non-hub hotspots, and non-hub coldspots. The results indicate that the high-amplitude regions are not necessarily highly connected hubs. Furthermore, the Broca aphasics utilized different hothub sets for the naming task. Both groups had dark functional networks composed of coldhubs. Thus, source amplitude–functional connectivity relationships could help reveal functional reorganizations in patients. The amplitude–connectivity combination provides a new perspective for pathological studies of the brain's dark functional networks.

Corresponding author
Zhong-Li Jiang,
jiangzhongli@njmu.edu.cn

## INTRODUCTION

In functional brain-imaging studies, there are two strategies can be utilized to identify a brain region as a responsible position for a certain task (e.g., picture naming) or event (e.g., seizure) (*Youssofzadeh & Babajani-Feremi, 2019*; *Van den Heuvel & Sporns,*

*2013*; *Daffertshofer & Van Wijk, 2011*; *Michel & Brunet, 2019*; *Biswal, 2012*; *Raichle, 2006*; *Salmelin et al., 1994*). The first strategy utilizes source localization to, for example, find activations that can be quantified by measurements, such as blood-oxygen-level-dependent signals detected through functional magnetic resonance imaging (fMRI) or current densities estimated via electroencephalography or magnetoencephalography (MEG) (*Bassett, Khambhati & Grafton, 2017*; *Bassett & Sporns, 2017*; *De Vico Fallani et al., 2014*; *Lewis, 2009*). The second strategy utilizes connectivity estimation to, for example, calculate connections that represent statistical dependencies, such as synchronizations or causal relationships among brain regions (*Farahani, Karwowski & Lighthall, 2019*; *Medaglia, 2017*; *Bassett & Sporns, 2017*; *Friston, 2011*; *Lewis, 2009*; *Sporns, 2002*). It is possible to detect statistical dependencies among regions regardless of whether the regions are highly activated. This yields an inevitable but not fully understood question regarding the relationship between "amplitude" and "connectivity".

In functional brain imaging, brain regions having large amplitudes are usually exhibited as hotspots, and other low-amplitude regions are considered coldspots. Considering this visualization tradition, *Van den Heuvel & Sporns (2013)* raised a question of whether or not the highly connected functional hubs were potential hotspots for developing diagnostic biomarkers or therapeutic targets. If we view the relationships between amplitude (i.e., hotspots and coldspots) and connectivity (i.e., hubs and non-hubs), there are four possible types of brain regions, including hothubs, coldhubs, non-hub hotspots, and non-hub coldspots. *Hassan et al. (2015)* reported that the deletion of regions of interest (ROI) having no more than 50% of the highest activations removed the temporal lobe from the brain networks of picture naming. Their findings implied that there were potential coldhubs in functional brain networks and that the possible contribution of "low-energy" regions should be further studied.

To the best of our knowledge, the relationship between activation and functional connectivity remains unexplored regarding task-related functional brain imaging. Few studies have investigated whether highly activated regions are also hubs in the task-related brain network. There are also limited reports of the pathological changes of activation-connectivity relations, and whether non-trivial activation-connection relations exist. In this study, we hypothesize that activation–connection coupling patterns exist for the performance of a naming task (i.e., significantly negative or positive correlations by healthy persons). We also propose that persons having post-stroke aphasia also have uncoupling patterns or opposite coupling patterns compared with healthy persons. The term "coupling" means that there is a significant correlation. The term "uncoupling" means there is a non-significant correlation. The term "opposite coupling" means that both correlations are significant, but their coefficients have a different sign. Based on a picture-naming task requiring MEG scanning, we investigate whether the highly connected regions are associated with low or high amplitudes. This allows detection of regions having both high activations and critical positions in brain networks (viz., hothubs). Moreover, we also explore the subnetworks comprising the coldhubs. Because such subnetworks are shown as dark parts in functional brain imaging, we call them dark functional networks.

## MATERIALS & METHODS

Five male persons having Broca's aphasia were recruited from the Rehabilitation Medical Center of the First Affiliated Hospital of Nanjing Medical University. The inclusion criteria were as follows: (1) a single left-hemisphere stroke confirmed by computed tomography or magnetic resonance imaging (MRI); (2) Broca's aphasia determined by the Mandarin version of the Western Aphasia Battery (WAB); and (3) native right-handed speakers of mandarin Chinese. The exclusion criteria were as follows: (1) severe vision or hearing impairment; (2) any neurological or psychiatric complications other than stroke, including but not limited to the diagnosis of mild cognitive impairment or dementia before stroke; and (3) contradictions with MRI testing. Five healthy volunteers (four males and one female) were enrolled in the control group. All participants performed the picture-naming task in a single session. The demographics are listed in Table 1. All participants signed an informed consent form that was approved by the Ethics Committee of The First Affiliated Hospital of Nanjing Medical University (No.2011-SRFA-025). All examinations were carried out under the guidance of the Declaration of Helsinki. Before the MEG test, all participants completed the WAB, which includes spontaneous speech, auditory comprehension, repetition, and naming tasks. The WAB suggests aphasia diagnosis for an aphasia quotient (AQ) <93.8. The WAB results can also be used to categorize the aphasia into different subtypes, such as Broca's or Wernicke's aphasia. All stroke participants in our study were diagnosed with Broca's aphasia.

### Experiment paradigm

The participants lay supine on the MEG scanner bed. A set of pictures was randomly presented onto a screen for the naming task. The stimuli consisted of 40 black-and-white outline drawings presented on a white background (see File S1). They were from the 100 words of the Kent–Rosanoff word association test (*Kent & Rosanoff, 1910*). All selected stimuli were a picturable noun or had a noun part of the meaning. They were both emotionally neutral and with high frequency (*Wang, Bing & Hou, 2010*). We modified a previously reported delayed naming task (*Laganaro et al., 2008*) based on the paradigm of *Levelt et al. (1998)*. Before the experiment, participants were trained to name aloud each picture correctly. This was done to confirm that the participants could perform the naming task in the scanning session (*Ellis et al., 2006*). Each trial comprised the following procedures (Fig. 1): a picture was shown on the screen for 200 ms, followed by a blank white screen for 1,000 ms. Next, a white question mark on a black background was shown for 2,000 ms. This was followed by a 1,000 ms black interval preceding the next trial. During the target-picture presentation, the participants were expected to perform overt naming. The question mark was a warning for the participants, indicating that they would soon be expected to perform overt naming. The 40 stimuli were randomly arranged as a sequence. The original design of this study was to let the participants do three runs in each experiment, and each run completed the sequence. However, our preliminary test showed that the patients could not tolerate a long test procedure, particularly when they were required to stay alone in a closed room without any tasks, waiting for the next run. We finally allowed all the participants to execute only a run of 100 trials (i.e., two-and-a-half sequences). There were no close

Lin et al. (2020), *PeerJ*, DOI 10.7717/peerj.10057

**Table 1 Demographic characteristics.**

| | Subjects | B1 | B2 | B3 | B4 | B5 | C1 | C2 | C3 | C4 | C5 |
|---|---|---|---|---|---|---|---|---|---|---|---|
| | Gender (Male/Female) | Male | Male | Male | Male | Male | Female | Male | Male | Male | Male |
| | Handedness (R/L) | R | R | R | R | R | R | R | R | R | R |
| | Age at onset (years)* | 71 | 50 | 48 | 55 | 19 | 68 | 55 | 55 | 54 | 70 |
| | Education duration (years)** | 7 | 10 | 10 | 16 | 9 | 8 | 8 | 9 | 12 | 9 |
| | Time after onset (months) | 8 | 5 | 5 | 42.7 | 9.4 | – | – | – | – | – |
| | Stroke type | hemorrhage | hemorrhage | infarction | hemorrhage | infarction | – | – | – | – | – |
| | Lesion locations | left (F, T, P, O) | left (basal ganglia) | left (F, T, P) | left (basal ganglia) | left (F, T, P) | – | – | – | – | – |
| | AQ*** | 31.4 | 64.6 | 43.8 | 53.5 | 58.3 | 98 | 99.4 | 99.4 | 99.6 | 99.8 |
| | Spontaneous speech | 6 | 12 | 8 | 10 | 12 | 19 | 20 | 20 | 20 | 20 |
| WAB | Auditory comprehension | 90 | 138 | 154 | 129 | 133 | 200 | 194 | 200 | 198 | 198 |
| | Repetition | 29 | 70 | 49 | 64 | 57 | 100 | 100 | 97 | 100 | 100 |
| | Naming | 23 | 54 | 13 | 39 | 48 | 100 | 100 | 100 | 100 | 100 |

**Notes.**

AQ, aphasia quotient; B, persons with Broca's aphasia; C, control subjects; F, frontal lobe; T, temporal lobe; P, parietal lobe; O, occipital lobe; WAB, Western Aphasia Battery.

T-tests between control group and Broca's aphasics with * $P = 0.30$; ** $P = 0.24$; *** $P = 0.00092$.

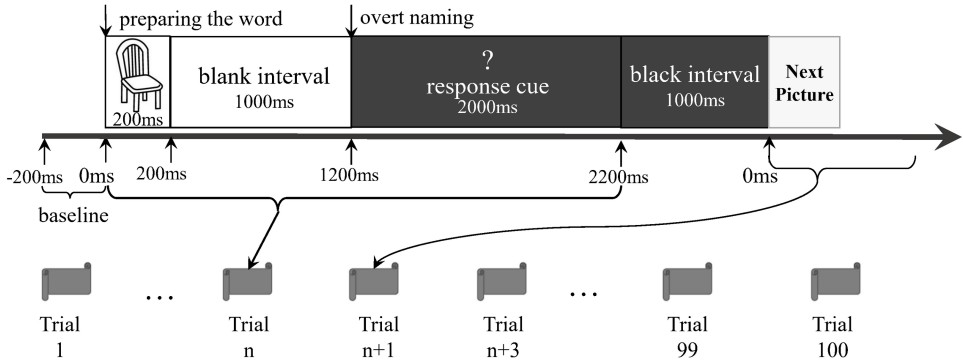

**Figure 1** Experiment paradigm.

distances of the same pictures in the run. The voices were transmitted outside the shielded room via a plastic tube and were recorded by researchers. The records were then classified as correct responses (i.e., target word or suitable alternatives) and incorrect responses (i.e., no-response or unsuitable words), according to the works of Feng Lin, Shao-Qiang Cheng, and Dong-Qing Qi. Only the consensuses would then become the final decisions. Then, the trials having correct responses were selected for further analyses.

## Data acquisition

Using a CTF-275 whole-head MEG scanner (VSM Canadian Medical Technology Company), available from the MEG Center of the Affiliated Brain Hospital of Nanjing Medical University, raw files of magnetic fields were recorded within the 0.03–100-Hz band. The sample rate was 1,200 Hz. The participants lay relaxed with their heads fixed under the sensory matrix. They were trained to avoid limb movements, head movements, and frequent blinking during the test. The pictures were delivered in a pseudorandom sequence using BrainX6.0 (Cincinnati Children's Hospital Medical Center, OH, USA) (*Dinga et al., 2018*; *Xiang et al., 2001*). The pictures were projected through an aperture onto a mirror, which, in turn projected the beam onto a screen. The horizontal and vertical visual angles were 3–4° and 1–2°, respectively. The space between the screen and the nasion of each participant was adjusted to a comfortable distance of 35–45 cm. The head positions were measured at the beginning and end of each session. Trials exhibiting motions greater than 0.5 cm were excluded.

After the MEG test, the participants underwent an MRI session for lesion delineation and source localization. The MRI data were acquired on a Signa (GE Medical Systems, USA) 1.5-T scanner capable of producing a high-resolution T1-weighted anatomical volume image (TR: 33 ms; TE: 9 ms; recording matrix: 256 × 256 pixels; excitation: 1; the field of view: 240 mm; and slice thickness: 1.4 mm). Transformation of the MEG coordinate system into MRI-defined space was achieved with the aid of three fiduciary points marked on the nasion and the right and left pre-auricular points with MRI markers, which were identical to the positions of the coils used in the MEG.

## Data processing

Brainsuite18a was used for MRI segmentation. Cortical surfaces were extracted from the T1 MRI data. The surface and volume registration was established based on the USCBrain Atlas, which has a total of 65 cortical regions per hemisphere (i.e., the cortex has 130 **original scouts**) (see Table S1) (*Joshi et al., 2017*). The anatomic data, including the MRI results and the registered surface files with their corresponding MEG data, were imported into Brainstorm (*Tadel et al., 2019*; *Baillet, 2017*; *Tadel et al., 2011*). The recordings were preprocessed using DC-offset correction, linear trend removal, 1–45-Hz band-pass filtering, and 50-Hz notch filtering. Artifacts were detected via signal-space projection and independent component analysis. The raw MEG files were epoched from −200 to 600 ms with a reduced sample rate of 600 Hz. As shown in Fig. 1, the epoched time window contained the word generation stages rather than the overt speaking actions. The pronouncing periods were only used to ensure that the participants paid attention to and were involved in the task. Although the control participants produced correct responses for all trials, the aphasics had incorrect responses. After discarding the bad trials having incorrect responses or unremovable artifacts, we uniformly resampled the trails into 60 correct ones for each participant for further analyses of the source-space functional connectivity. Although there were repetitions of some pictures, all participants provided corresponding trails during the run. Source reconstruction was performed using weighted minimum-norm estimation (WMNE) on a current density map and constrained dipole orientations (*Hincapié et al., 2017*). Each original scout was subdivided into six sub-regions. The final subdivided atlas had 776 **refined scouts**. Trials were averaged on the source level for each person using the weighted arithmetic average. The source data for each person were projected onto the USCBrain_BrainSuite_2017 template. We further averaged the projected source files on the group level, finally obtaining one source file per group. Following the previous work of *Hassan et al. (2015)*, we extracted six picture-naming stages: t1 (0–119 ms, visual feature extraction); t2 (120–150 ms, object recognition); t3 (151–190 ms, memory access); t4 (191–320 ms, semantic processing); t5 (321–480 ms, phonological encoding); and t6 (481–535 ms, articulation). Phase-locking values (PLV) among the 776 refined scouts were estimated for each stage on the gamma frequency band of 30–45 Hz. As reported by *Hassan et al. (2015)*, to reduce confounding factors from correlations of adjacent regions, all PLVs were normalized by the mean and standard deviation matrix of the baseline (−200 ms–0 ms) time-window.

## Graph modeling

We use graph theory terminology (*De Nooy, Mrvar & Batagelj, 2018*) and the expressions from the work of *Rubinov & Sporns (2010)* for brain network analyses. A node is a scout of the brain region, and an edge is an undirected line. Size refers to the number of nodes in a network. A cluster refers to a subnetwork of the entire network. In this study, the nodes were the 776 refined scouts, and the edge weights were normalized PLV scores larger than 1.96. We established six undirected refined networks on the subdivided USCBrain Atlas for each person. These personal networks were utilized for further within-subjects and inter-group permutation t-tests. We also established refined networks on the

averaged source files for each group. These averaged original networks were applied for further hothub and coldhub detection. The amplitude of each refined scout was measured according to the estimated electric density in a physical unit of picoampere. The scouts were partitioned using a functional scheme (see Table S1) that included eight systems (*Muldoon et al., 2016*; *Gu et al., 2015*): (1) frontoparietal, (2) medial default mode, (3) visual, (4) ventral temporal association, (5) attention, (6) cingulo-opercular, (7) motor and somatosensory, and (8) auditory. By partitioning the functional systems into the left and right hemispheres, the final scheme contained 16 functional modules. In our descriptions, the term "strongly connected" indicates the weight strength of connections. The term "highly connected" is known as the number of connections. The term "heavily connected" implies to the measures concerning both weights and numbers of connections. The term "highly activated" identifies the amplitude of a node.

## Network thresholding

To find nontrivial functional connections and get robust findings, *Hassan & Wendling (2018)* suggested the reduction of the edges in a graph using a series of threshold values. In this work, we utilized the island decomposition technique embedded in Pajek 5.07 to perform fast thresholding operations (*Mrvar & Batagelj, 2016*). Pajek is a network analysis software package that can achieve fast speed and memory-efficient calculations for large and densely structured graphs (*Pavlopoulos et al., 2017*). The island decomposition defines an island as a cluster with highly weighted within-cluster edges. In particular, the weights inside the cluster must be larger than the weights to neighboring nodes outside the cluster. Pajek called for two inputs to control the island thresholding processes: the minimum and maximum island sizes. In this study, the former was fixed to three. This is the minimum graph size that can support calculating clustering coefficients. We shifted the maximum size from 50 to 750 in increments of 50. Given the half nodes of each hemisphere and the total nodes of the entire network, we also set 388 and 776 as candidate maximum island sizes. These numbers determined the possible size of the main components in the obtained networks. At each naming stage, the island decomposition generated a series of m-island graphs (where m denotes the maximum island size).

## Graph measures

The refined networks and their thresholded networks were weighted undirected graphs lacking loops and multiple edges. Based on these graphs, we calculated nine variables for each node (*De Nooy, Mrvar & Batagelj, 2018*; *Sporns, 2016*; *Rubinov & Sporns, 2010*).

The first six variables were calculated without utilizing the network partitions of the 16 functional modules. The **degree centrality** of a node in its weighted definition is the sum of the edge weights connected to the node. This is a measure of the strength of the node in the range of its firstly connected neighbors. The largely weighted degree centrality indicates that the node is heavily connected. The **betweenness centrality** of a node in its weighted definition is the number of weighted shortest paths going through it. This is a measure of the key connector role of the node in transferring information or its role as a bottleneck in blocking information flowing through the graph paths. **Transitivity** is the probability that

the first-connected neighbors of a node are also connected. *Barrat et al. (2004)* reported that the weighted definition of transitivity was the local clustering coefficient having constant edge weights between the target and adjacent nodes. The **k-coreness** of a node was derived from a thresholding method called the "$k$-core decomposition". This method reduces the graph to a maximal subgraph in which each node has at least a degree, $k$. A valid $k$-coreness means that a node belongs to the $k$-core but not to the $(k+1)$-core, and the $k$-coreness measures whether a node involves the highly connected core of the brain graph. The **Laplacian centrality** is a measure that concerns the possible destructive effects of deactivating or deleting a node from a graph (*Qi et al., 2013*). The higher the Laplacian centrality of a node, the more indispensable it is. Note that the $k$-coreness and Laplacian centrality algorithms do not consider the edge weights. Finally, the **eigenvector centrality** of a node is assigned based on whether the node connects to many other nodes and/or to highly connected nodes. Highly scored nodes are highly connected with highly connected neighbors. That is, they are hubs of the graph (*De Nooy, Mrvar & Batagelj, 2018*). Weighted definitions were applied to the eigenvector centralities obtained in this study.

To measure the distribution of a node's connections across modules, we also calculated three variables that depend on the network partitions. We used 16 functional modules, as detailed in Table S1. The **participation coefficient** of a node measures the distribution of its inter-modular connections. If a node is only connected to nodes in the same module, the participation coefficient is zero. If the node is equally connected to all other modules, the participation coefficient is one. The **gateway coefficient** of a node refers to both its inter-modular and within-modular connections (*Vargas & Wahl, 2014*). If a node links to the hub within its module and occupies most of the outer connections from other modules to its module, this node has a larger gateway coefficient. As described by *Vargas & Wahl (2014)*, this coefficient makes it feasible to identify nodes with unique inter-modular connections. The **within-module-degree z-score** of a node is its normalized number of edges that connect to other nodes in the same module of the target node.

In this work, all graph measures were calculated using the undirected definitions (*Hassan & Wendling, 2018*; *Rubinov & Sporns, 2010*). The degree, betweenness, transitivity, $k$-coreness, and eigenvector centrality were calculated using the igraph 1.2.4.1 package (*Csardi & Nepusz, 2006*) of the R software. The Laplacian centrality was calculated using Pajek 5.07 (*Mrvar & Batagelj, 2016*). The within-module-degree z-score, participation coefficient, and gateway coefficient were obtained using the brainGraph 2.2.0 package (*Watson, 2019*) of the R software.

## Hub and hotspot detection

The $z$-scores of the amplitudes were calculated for both subdivided and original scouts by comparing the baseline of $-200$ ms. At each stage, the refined scouts were categorized into hotspots or coldspots depending on whether their amplitudes were larger than the mean plus standard deviation (*Hassan et al., 2015*). The network hubs were detected using Pajek 5.07. As defined by *De Nooy, Mrvar & Batagelj (2018)*, hubs are nodes with top-level eigenvector centralities. However, there is no optimal strategy to determine the top-level cutoff value. In this study, the number of hubs was set to the number of hotspots

in each graph. This is an arbitrary selection that includes 11.86%–16.37% nodes. This strategy followed a rule of thumb (*De Nooy, Mrvar & Batagelj, 2018*) and did not exceed the arbitrary threshold of 30% (*Youssofzadeh & Babajani-Feremi, 2019*). Finally, all nodes were categorized into four types: **hothubs**, **coldhubs**, non-hub coldspots, and non-hub hotspots.

## Statistical analysis

Figure 2 shows the data-processing workflow. Inter-group permutation t-tests for the amplitude z-scores on the 130 scouts at each naming stage were processed using Brainstorm (*Tadel et al., 2019*; *Tadel et al., 2011*), with $p < 0.05$ as the level of significance. Within-subject Pearson coefficients were calculated between the amplitude (i.e., electric densities in a unit of picoamperes) and each of the graph measures on the refined 776 scouts. These were partial correlations with the effect of subjects removed. Thus, the individuals were regarded as a third variable that should be adjusted when comparing the amplitude with a graph measure. The 95% confidence intervals and significances of the Pearson coefficients were estimated using the psych 1.8.12 package (*Revelle, 2019*) of the R software. To explore the entire continuum of m-island graphs and select an optimal m value, we plotted all coefficients on a coordinate system for which the $x$-axis denoted the maximum island size (see Figs. S1 and S2). The left pole included m-island graphs having small but strongly connected clusters (i.e., the "rich-club" structures). The right pole contained m-island graphs having both large and highly connected clusters (i.e., large island structures with a wide range of edge weights from weak to strong). If a correlation were significant ($p < 0.05$), it could be regarded as a **coupling correlation**. Conversely, a non-significant correlation could be termed an **uncoupling correlation.** We also did a permutation test on inter-group correlation coefficients with 1,000 randomizations by shuffling the group assignments of individuals [i.e., the 10 individuals in a permutation were randomly reassigned into two groups (5 ones per group) without replacement]. For each permutation, an inter-group difference value of partial correlation coefficients were calculated. The 1,000 values formed an estimation for the distribution of inter-group differences. The statistics of correlation coefficients followed the estimation approach with 95% confidence intervals (*Calin-Jageman & Cumming, 2019*). The brain networks were visualized using Pajek 5.07 (*Mrvar & Batagelj, 2016*) and VOSviewer 1.6.13 (*Van Eck & Waltman, 2009*).

## RESULTS

### Amplitude–connectivity analysis

Figures 3 and 4 show the Pearson correlation coefficients between the ROI amplitudes and their graph measures, for measures independent of and dependent on the brain modules, respectively. A positive significant coefficient suggests that the regions having higher activation levels tend to have high values of the corresponding graph measure. That is, hotter spots occupy pivotal network positions. Conversely, a negative significant coefficient suggests that the regions having lower activation levels tend to have high values of the corresponding graph measure (i.e., colder spots are positioned at important network locations). Figure 3 shows coefficients in m776-islands, and Fig. 4 shows coefficients in

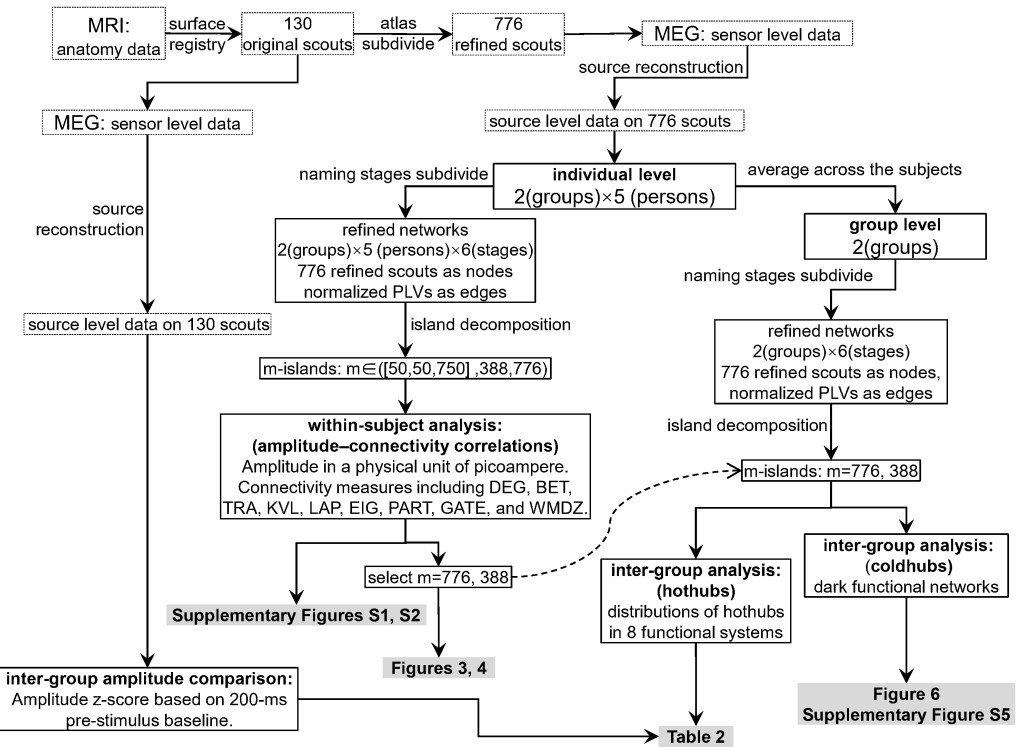

**Figure 2  Data processing workflow.** The dashed curved arrow denotes that the final reported m776 and m388 islands are selected from a series of m values by interpreting the Figs. S1 and S2. MEG, magnetoencephalography; MRI, magnetic resonance imaging; DEG, weighted degree; BET, weighted betweenness; TRA, weighted transitivity; KVL, k-value of coreness; LAP, Laplacian centrality; EIG, eigenvector centrality; PART, participation coefficient; GATE, gateway coefficient; WMDZ, within module degree *z*-score.

m388-islands. The selection of m values in the two figures was determined by methods summarized in Figs. S1 and S2. There are two positions on the *x*-axis that optimally separate the inter-group distribution of coefficients (see vertical dashed lines in Figs. S1 and S2). Figures S3 and S4 support the significant inter-group differences of Pearson coefficients by showing the 95% confidence intervals of 1,000 random permutations.

Figures 3 and 4 provide two obvious findings. First, the healthy controls had positive significant coefficients, especially at t4, t5, and t6 (Fig. 3). Second, the coefficients of the two groups deviated from each other with two types of deviation. For Type-I, the two groups had significant but opposite coefficients (i.e., there was a deviation between the two coupling correlations, and the degree results for t4 and 5 are shown in Fig. 3). For Type-II, one group had significant coefficients, but the other group showed no correlation significance (i.e., the coupling and uncoupling correlations differed, and the degree results for t1, t3 and t6 are shown in Fig. 3). For the Broca-group view results, Type-I means that significant differences in amplitude–connectivity patterns were noted between the Broca aphasics and the control group, and the Type-II deviation implies emergence or vanishing of amplitude–connectivity patterns in the Broca group.
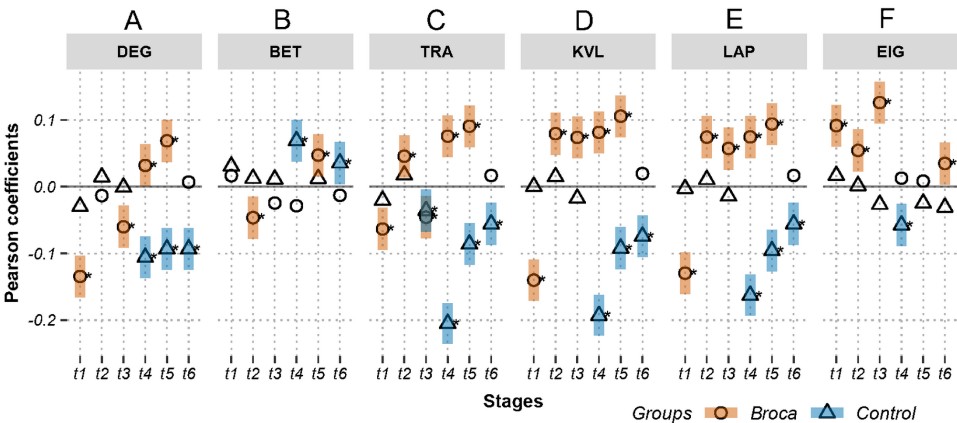

**Figure 3** **Relations between activations (picoampere) and module-independent graph measures in m776-islands.** Activations are the estimated electric densities (in a physical unit of picoampere) in the source space. The partial Pearson coefficients between the activations and graph measures are calculated for m776-islands at different stages. The significant coefficients ($p < 0.05$) are marked with asterisks. The 95% confidence intervals having significant coefficients are marked by transparent colored bars. DEG, weighted degree; BET, weighted betweenness; TRA, weighted transitivity; KVL, k-value of coreness; LAP, Laplacian centrality; EIG, eigenvector centrality; t1: 0–119 ms, visual feature extraction; t2: 120–150 ms, object recognition; t3: 151–190 ms, memory access; t4: 191–320 ms, semantic processing; t5: 321–480 ms, phonological encoding; t6: 481–535 ms, articulation. A positive coefficient marked with an asterisk denotes that strongly activated brain regions are more likely to be highly connected hubs. A negative coefficient marked with an asterisk suggests that highly connected hubs are more likely to be with weak intensities of activation. The separation of the confidence intervals having opposite values of coefficients infers that the two groups have significantly different amplitude–connectivity relationships. One significant correlation having another nonsignificant correlation also implies that there are interconditional differences of amplitude–connectivity relationships.

In Fig. 3, Stages t4 (semantic processing) and t5 (phonological encoding): the degree, transitivity, $k$-coreness, and Laplace show Type-I deviation with positive significant coefficients in the Broca group. Type-I deviations having positive coefficients in the Broca group occurred at t1 of the eigenvector, t2 of the transitivity, $k$-coreness, Laplace, and eigenvector, t3 of the $k$-coreness, Laplace, and eigenvector, t5 of the betweenness, and t6 of the eigenvector. Type-I deviations having negative coefficients in the Broca group occurred at t1 and t3 of the degree, t1 of the transitivity, $k$-coreness, and Laplace. Type-I deviations having positive coefficients in the control group occurred at t4 and t6 of the betweenness. Type-I deviations having negative coefficients in the control group occurred at t4 of the eigenvector, and t6 of the transitivity, $k$-coreness, and Laplace. In Fig. 4, within-module-degree z-score shows no significant correlations for each group. The t1, t2, and t3 show significant positive correlations in the Broca group. The t3, t4, t5, and t6 show a significant negative correlation in the control group.

## Hotspot and hothub analysis

Although there were different possible categories for the six subdivided scouts in the same 130 original ones, we reported the hothubs on the level of the original 130 scouts (Fig. 5) if there was at least one hothub on the level of the 776 subdivided scouts. This

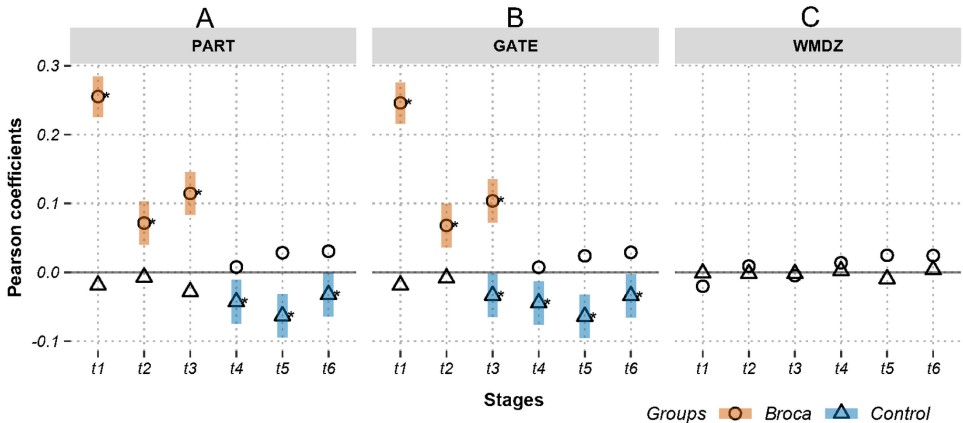

**Figure 4** **Relations between activations (picoampere) and module-dependent graph measures in m388-islands.** Activations are the estimated electric densities (in a physical unit of picoampere) in the source space. The partial Pearson coefficients between the activations and graph measures are calculated for m388-islands at different stages. The significant coefficients ($p < 0.05$) are marked with asterisks. The 95% confidence intervals having significant coefficients are marked by transparent colored bars. PART: participation coefficient; GATE: gateway coefficient; WMDZ: within module degree z-score; t1: 0–119 ms, visual feature extraction; t2: 120–150 ms, object recognition; t3: 151–190 ms, memory access; t4: 191–320 ms, semantic processing; t5: 321–480 ms, phonological encoding; t6: 481–535 ms, articulation. A positive coefficient marked with an asterisk denotes that strongly activated brain regions are more likely to be highly connected hubs. A negative coefficient marked with an asterisk suggests that highly connected hubs are more likely to be with weak intensities of activation. The separation of the confidence intervals with opposite values of coefficients infers that the two groups have significantly different amplitude–connectivity relationships. One significant correlation with another nonsignificant correlation also implies that there are interconditional differences of amplitude–connectivity relationships.

was because most previous reports did not interpret functional brain activities on a resolution as did the level of the 776 refined scouts (as shown in Table S2). Therefore, Fig. 5 reports the regions in which hothubs occurred. We also marked eight functional systems in this figure using different colors. As detailed in Fig. 5, the Broca group scores for InfOcciGyr_VenPst_R (right inferior occipital gyrus ventroposterior), PreCune_Sup_R (right precuneus superior), and MidOcciGyr_DsoAnt_L (left middle occipital gyrus dorsal anterior) had significantly lower amplitude z-scores (cells with superscript a). However, they remain in the list of hothubs. Compared with the control group, the Broca group did not use the cingulo-opercular hothubs (yellow cells) across all stages, and the hothubs in the medial default mode (green cells) were absent at t3. They required more visual system regions at each stage (white cells) and resorted to the attention system in the last three stages (pink cells). For the control-group t4 and the Broca-group t5, most hothubs were from the frontoparietal system.

## Brain network visualizations

We visualized the m776 islands by showing their hotspots and hubs. Figure 6 provides a visualization of the brain networks for the semantic processing stage of t4. This figure presents two aspects of the brain networks by changing the node size according to both amplitude and linking strengths. Four colors are used in this figure: red (hothubs), green

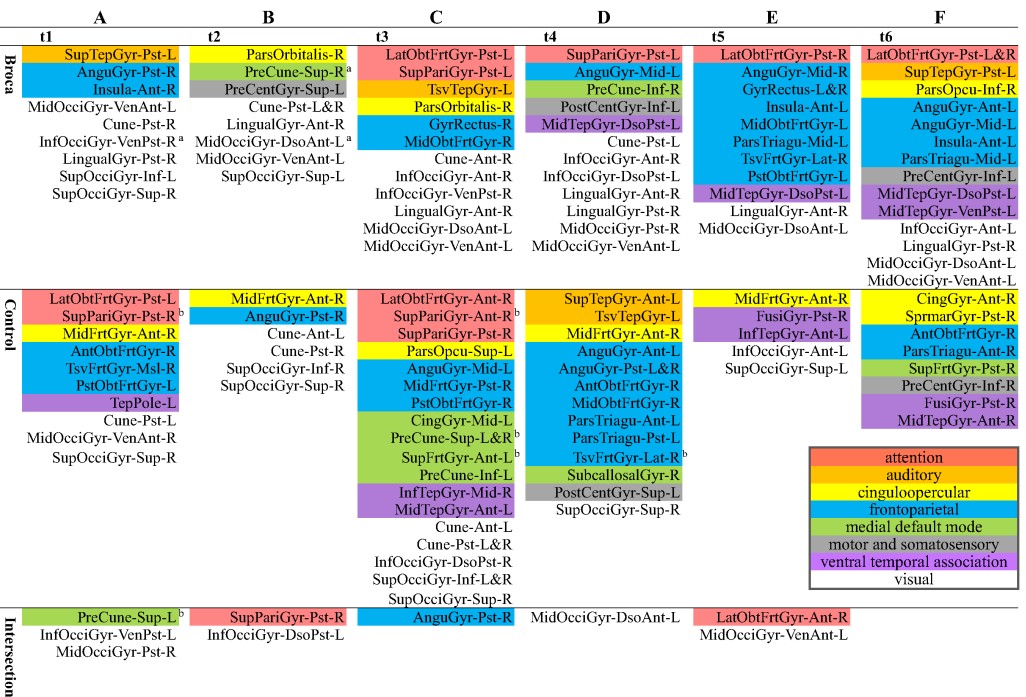

**Figure 5 Hothubs in m776-islands.** [a] Significantly lower $z$-scores for amplitudes of Broca group than those of the control group (permutation $t$-test, 1,000 randomizations). [b] Significantly higher $z$-scores for amplitudes of control group than those of Broca group (permutation t-test, 1,000 randomizations). (A) t1(0–119 ms, visual feature extraction): InfOcciGyr_VenPst_R ($p = 0.016$), SupPariGyr_Pst_R ($p = 0.008$), PreCune_Sup_L ($p = 0.02$). (B) t2 (120–150 ms, object recognition): PreCune_Sup_R ($p = 0.046$), MidOcciGyr_DsoAnt_L ($p = 0.004$). (C) t3 (151–190 ms, memory access): SupPariGyr_Ant_R ($p = 0.016$), PreCune_Sup_L ($p = 0.024$), PreCune_Sup_R ($p = 0.017$), SupFrtGyr_Ant_L ($p = 0.039$). (D) t4 (191–320 ms, semantic processing). (E) t5 (321–480 ms, phonological encoding). (F) t6 (481–535 ms, articulation). Colors for functional modules: Pink: attention; orange: auditory; yellow: cinguloopercular; blue: frontoparietal; green: medial default mode; grey: motor and somatosensory; lilac: ventral temporal association; white: visual.

(coldhubs), yellow (non-hub hotspots), and blue (non-hub coldspots). The full set of visualizations is provided in Fig. S5.

## DISCUSSION

### Thinking about amplitude, connectivity, hothubs, and coldhubs

In the last two decades, a new trend known as functional brain network analysis, which involves identification of critically connected positions as specific regions for a certain task or event, has been developed (*Farahani, Karwowski & Lighthall, 2019*; *Medaglia, 2017*; *Bassett & Sporns, 2017*; *Lewis, 2009*; *Sporns, 2002*). The new principle based on connectivity challenges the traditional paradigm that identifies critical regions by their levels of activation/amplitude. It is thus necessary to clarify whether or not a highly/heavily connected region also has a high activation level. Although there is a growing number of reports about either or both of the two aspects, few studies have considered the
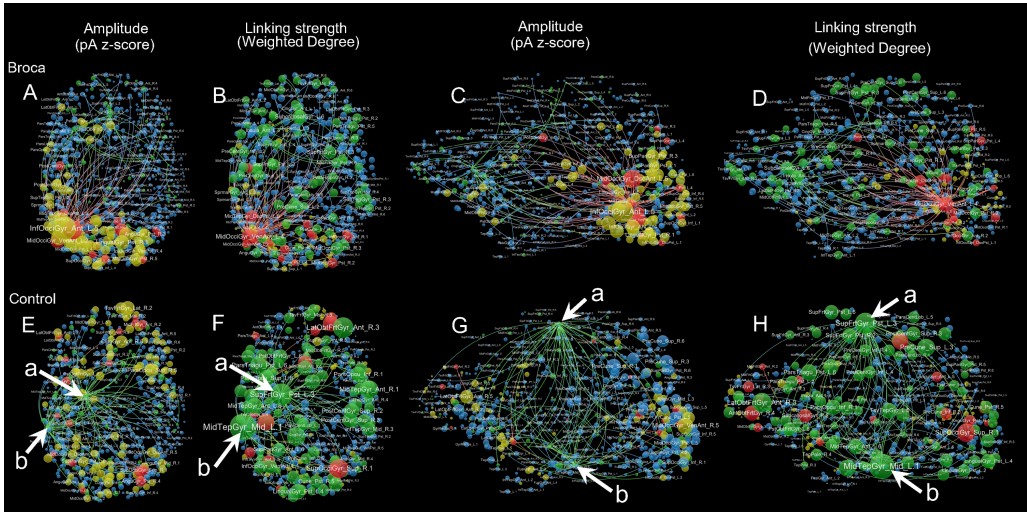

**Figure 6 Sample amplitude and linking strength distributions in t4.** The t4 is the semantic processing window of 191–320 ms. There are four node types: hothubs (red), coldhubs (green), non-hub hotspots (yellow), and non-hub coldspots (blue). Hotspots are identified based on their amplitudes that exceeded the mean-plus-one standard deviation. Hubs are identified based on their eigenvector centralities as defined in Pajek, with the same number of hotspots at each stage. The top-weighted 100 edges are plotted, and the edges are colored according to their terminals. For the amplitude weighted layouts, node areas are in proportion to the z-scores of their amplitudes, including top-down view (A) and left lateral view (C) of the Broca group, and top-down view (E) and left lateral view (G) of the control group. For the linking strength weighted layouts, node areas are in proportion to the total linking strengths (i.e., weighted degrees), including top-down view (B) and left lateral view (D) of the Broca group, and top-down view (F) and left lateral view (H) of the control group. The amplitude weighted layouts are remarkably different from the linking strength weighted layouts, suggesting that it is necessary to reconsider the role of coldhubs in the functioning brain. Arrow [a] Left SupFrtGyr_Pst (left superior frontal gyrus posterior). Arrow [b] Left MidTempGyr_Mid (left middle temporal gyrus middle).

relationship between them. By using resting-state and finger tapping fMRI tests, *Zhou et al. (2014)* explored the fractional amplitudes of the lower-frequency fluctuations and the thalamic seed-based functional connectivity in persons affected by mild traumatic brain injuries. Although these researchers did not provide quantitative measurements for the amplitude–connectivity relations, their side-by-side comparison revealed co-occurring amplitude and connectivity decreases. This finding suggests that the target patients had attenuated thalamocortical networks (*Zhou et al., 2014*). Furthermore, based on resting-state fMRI tests, *Zhang et al. (2015)* defined the amplitude–connectivity coupling strength as the correlation coefficients between the amplitude of the low-frequency fluctuation and the functional connectivity density. Compared with healthy controls, the amplitude–connectivity coupling strengths in persons having mesial temporal lobe epilepsy were found to be significantly lower in the mesial temporal structures and significantly higher in the default-mode regions. This suggests that the amplitude–connectivity uncoupling pattern can be used for differentiating patients with healthy controls. Base on a picture-naming test via MEG scanning, the present study investigated the task-state amplitude–connectivity correlation. We expanded the field by answering the following two questions.

First, do amplitude–connectivity relationships exist across scales? This question considers whether amplitude–connectivity relationships occur in different spaces and at different times. For the second aspect, we utilized a MEG naming test with a six-stages of separation (see Figs. 3, 4 and 5) (*Hassan et al., 2015*). For the first aspect, we adopted island decomposition (*De Nooy, Mrvar & Batagelj, 2018*) as the thresholding method for brain graph modeling. This method establishes continuums for brain graphs and allows distributions of amplitude–connectivity patterns in the continuums to be found (see Figs. S1 and S2).

Second, how can the most important brain regions be identified? In the amplitude dimension, the hotspots having high activity levels are responsible for the task performance. In the connectivity dimension, the highly connected hubs are pivotal to signal processing. Based on both dimensions, we classified brain regions into four categories: hothubs, non-hub hotspots, coldhubs, non-hub coldspots. If the correlation of amplitude–connectivity is significant ($p < 0.05$) and strong (with coefficients approaching 1 or $-1$), there should be only hothubs or coldhubs. If the correlations are non-significant or significant with weak coefficients, there should be four types of nodes in functional brain networks, and the hothubs should be the most important position, because they match both amplitude and connectivity principles.

## Coupling and uncoupling amplitude–connectivity patterns in Broca and control groups

The coefficients were distributed in a similar pattern at different stages and with different graph measures (see Figures 3 and 4, and Figs. S1 and S2). For the Type-I deviations, the 95% confidence intervals of the two groups seldom overlapped along the island continuums. This suggests that, although both groups had amplitude–connectivity coupling relationships, their patterns were opposite. The Type-II deviations revealed the differences in the coupling and uncoupling patterns of the different groups. The controls tended to have negative values, whereas the Broca persons tended to have positive coefficients. Figure 3 also showed that the Broca group had more coupling relationships than did the control group. For example, the t2 and t3 of the $k$-coreness, Laplace, and eigenvector had Type-II deviations with positive correlations in the Broca group. These findings suggest that, at the object recognition (t2) and memory-access (t3) stage, the regions having high activation levels tended to be hubs (by eigenvector centrality) in core structures (by $k$-coreness), and they occupied critical positions (by Laplacian centrality) in the Broca aphasics. However, the interesting findings of our study were that the healthy controls had uncoupling or opposite patterns, compared with the Broca patients. This implies that the amplitude and connectivity of healthy persons are usually independent (as indicated by the uncoupling pattern) or negatively related to each other (as indicated by the coupling pattern with negative coefficients). We can thus theorize that a healthy brain follows the principle of least effort. That is, to perform a task without requesting more resources, the highly activated regions tend to run without unnecessary weak connections.

Close analysis of the coefficient distributions in Fig. 3 (and Fig. S1) yielded further detailed findings. Betweenness appears to be a measure that is usually uncoupled with

amplitude, especially in islands having small m-values (see Fig. S1), suggesting that the highly activated regions did not necessarily work to transport information between other regions. The transitivity shows mixed distribution patterns for the coefficients across all naming stages. As the transitivity measures the clustering possibility of a node's first neighbors, it seems that healthy persons tended to form highly clustered structures of low activation regions during t3 to t6 in islands having large m values (see Fig. 3 and Fig. S1). The Broca persons considered in this study exhibited the opposite distribution (i.e., the low activation regions had lower neighborhood clustering coefficients). This suggests that, during memory access (t3), semantic processing (t4), phonological encoding (t5), and articulation (t6), the low-activation regions in the patient's brain lost their internal organizations, and their highly excited regions formed a greater number of connections among their neighbors for task performance. That is, the impaired brain recruited more local-neighbor resources to perform the same tasks (*Medaglia, 2017*).

The weighted degree, $k$-coreness, and Laplacian centrality indicated negative coefficients in the m776 islands during t1 (see Fig. 3), but they showed positive coefficients in most m-islands (see Fig. S1). This suggests that weakly weighted edges (indicated by the m776 islands, see Fig. 3) in the Broca group supported low activation regions while being central (by the weighted degree), core (by the $k$-coreness), and critical (by the Laplacian centrality). However, most edges (in m-islands other than $m = 776$, see Fig. S1) allowed the highly activated regions to be central (by the weighted degree), core (by the $k$-coreness), and critical (by the Laplacian centrality). Moreover, the $k$-coreness and Laplacian centrality result from t2 to t5 show consistent forms of the positive coefficients (see Fig. 3), suggesting that most edges supported the highly activated regions while being core (by the $k$-coreness) and critical (by the Laplacian centrality). A similar pattern was obtained for the degree from t4 to t5, for the $k$-coreness and Laplacian centrality. Given the language production problem exhibited by the Broca persons, we can infer that these patients required more resources to reconstruct compensatory brain areas as task-specific regions. Figure 4 (and Fig. S2) supports this conclusion, because, among the Broca persons, the highly activated regions tended to participate in more functional systems during the first three stages.

## Hothubs interpretation

The hub detection method with participation coefficients should identify the optimal functional module partition scheme (*Muldoon et al., 2016*; *Van den Heuvel & Sporns, 2013*; *Gu et al., 2015*; *Medaglia, 2017*). The eigenvector centralities allow the detection of hubs in the network without considering partition schemes (*De Nooy, Mrvar & Batagelj, 2018*). To avoid the controversy regarding partition schemes, we used the eigenvector centralities to self-consistently define the network hubs. As shown in Fig. 3, the Broca-group, t1, t2, t3, and t6, had positive coefficients on the eigenvector centralities in m776-islands, suggesting that these patients tended to use highly activated regions as hubs. However, uncoupling distributions were apparent for t4 and t5, suggesting that the highly activated regions were not necessarily to be hubs in patients. In other words, the inference of hubs based on the region's activation level during the semantic processing and phonological encoding stage

of the naming task was uncertain for the Broca group. Interestingly, the control-group t4 stage had a negative correlation, implying significant coldhubs at this stage.

We identified the hothubs in the averaged brain of each group. The regions in Fig. 5 were both hotspots having amplitudes exceeding the mean-plus-one standard deviation and hubs having top-level eigenvector centralities for the same number of hotspots. First, the results suggest that there were only several intersections between the two groups. That is, the patients reorganized their functional brain networks using different sets of regions as their hothubs in response to the naming task. The hothubs in Fig. 5 were mostly reported as naming specific regions (as shown in Table S2). Note that some regions support cognitive functions related to naming tasks, such as visual recognition, long-term memory processing, and object configuration by the right posterior fusiform gyrus (*Song et al., 2019*). Second, the significantly attenuated regions, such as InfOcciGyr_VenPst_R (right inferior occipital gyrus ventroposterior) in t1, and MidOcciGyr_DsoAnt_L (left middle occipital gyrus dorsal anterior) and PreCune_Sup_R (right precuneus superior) in t2, also acted as hothubs in the Broca group. However, during the entire naming process, the Broca group used specific hothubs for which the activation levels did not differ from the control group. Third, both groups commanded more than one functional system at each stage. This suggests that naming should be a cross-modular task. Because our Broca participants were tested at least 5 months after their onset (see Table 1), we can infer that our observations were the results of functional reorganizations. The different hothubs and functional systems in each group (as detailed in Fig. 5) suggest that there were both intrasystem and intersystem reorganizations in the Broca group.

### Network visualizations and coldhubs exploration

For more detailed inspections of these reorganizations, we visualized each graph by categorizing the refined 776 regions into four types (as shown in Fig. 6 and Fig. S5). Figure 6 is an example of amplitude and linking-strength distributions in t4. Coldhubs are always present in each graph. For example, the left SupFrtGyr_Pst (superior frontal gyrus posterior, see Fig. 6 arrow a) and MidTempGyr_Mid (middle temporal gyrus middle, see Fig. 6 arrow b) are coldhubs occupying critical positions on the dorsal and ventral stream at the semantic processing stage. These coldhubs cannot be visualized by the traditional imaging strategy that depends on the scout activation level. Focusing on the linking-strength panels could reveal that the coldhubs (green nodes) exhibit a different functional brain network landscape. From these findings, a partial answer to the question raised by *Van den Heuvel & Sporns (2013)* on whether brain hubs are potential hot spots with clinical meaning could be derived. Network hubs are possible but not necessarily bound to be hot spots. Patients and their healthy controls differed both in the hothubs and in the coldhubs. The dark functional networks formed by coldhubs deserve more considerations in future studies.

### Methodological considerations, limitations, and clinical implications

Our study has several limitations. First, the connectivity modeling in MEG has spatial leakage problems (*Palva et al., 2018*; *Colclough et al., 2016*; *Brookes, Woolrich & Barnes,*

*2012*). PLV is a method lacking leakage correction (*Colclough et al., 2016*). *Palva et al. (2018)* recommended that researchers perform the full source-space interaction mapping rather than applying a seed-based approach. We utilized the connectivity strategy to interpret undirected graphs based on PLVs across the whole cerebral cortex. *Hassan et al. (2015)* previously used such graphs for studying picture-naming. They constructed an atlas with 1,000 scouts covering the whole cortex, calculated PLVs based on the source space estimated by the WMNE method, and normalized PLVs to reduce artefactual inflated connectivity. We constructed 776 scouts covering the whole cortex, calculated PLVs based on WMNE, and also normalized PLVs as connection weights. According to *Hincapié et al. (2017)*, if the interacting sources were extended patches having several dipoles in them, the minimum-norm estimation would provides better connectivity estimation than would the beamformer. We know that these considerations might not fully resolve the controversy about using PLV. This is an ongoing issue. *Rizkallah et al. (2020)* reported that PLV metrics of MEGs were significantly correlated with fMRI networks compared with other zero-lag corrected methods. Our report encourages us to explore more types of functional and effective connectivity in future works. For example, the phase-transfer entropy will establish directed graphs for detailed inspections of information flows in the brain (*Engels et al., 2017*).

Second, the estimated source values of the WMNE tended to be reduced with increasing depths, varying significantly between subjects. The Brainstorm's guides recommended using the current density map for computing connectivity measures. Although the coupling patterns explored by our study were based on within-subject analyses, we still reported findings based on the raw current density values in the unit of picoamperes. We also projected all source spaces into the same template to reduce the influences of inter-individual differences.

Third, we are aware of our small participant sample and the inconsistencies of stroke victims. One of the participants (B5) was significantly younger than the others. Although this is a possible confounding factor, the maximum variation strategy provides another view (*Patton, 2014*). If there any patterns were to emerge, despite the great variation, the patterns would have value in capturing the essential impacts of a program. Therefore, the amplitude–connectivity correlation patterns found in this study made sense for our original hypothesis. On the other hand, the effects of individuals were adjusted via partial correlation analysis. That was, we treated the individuals as the third variable when calculating the amplitude–connectivity correlation coefficients.

Fourth, the stroked structure should be considered in the study. There were brain volumes with encephalomalacia that could influence the results of source reconstruction. This is a technical problem without a satisfactory solution in the head-modeling and brain parcellation. The stroked-out areas remained part of the source space. Therefore, the results of stroke persons were estimates based on virtual structures of the brain. Nevertheless, the control group should not be influenced by this problem, and results from them should make sense for our hypothesis. Besides, the amplitude–connectivity relationships between groups remained unchanged after removing the stroked left hemisphere in both groups (see Figs. S6 and S7). Moreover, clinical applications of transcranial brain

stimulation (e.g., repetitive transcranial magnetic stimulation) usually inject electric currents into the brain by inversely assuming the source space of stimulation (*Biou et al., 2019*; *Norise & Hamilton, 2017*; *Coslett, 2016*). With the rapid expansion of non-invasive brain stimulation technology, the effective target region location is an issue strongly associated with clinical application. Therapists should decide to use one of two opposite strategies to execute stimulation (viz., to excite or inhibit a target position) (*Biou et al., 2019*; *Norise & Hamilton, 2017*; *Polania, Nitsche & Ruff, 2018*). Essentially, their decisions related to excitatory or inhibitory stimulation protocols are related to the hot or cold spots. The amplitude–connectivity correlation study reported herein presents a new perspective on this question by revealing the dark functional networks with coldhubs in the reconstructed source space. The effects of brain stimulation on the cold and hot hubs may differ and require more study. Further study should include more participants to confirm the cross-scale deviation patterns of the amplitude–connectivity relationships.

Apart from the perspective of brain stimulation, speech-language therapy is another field related to our findings. Picture-naming is a widely used therapy task in speech-language therapy. However, various variables can influence the outcomes of linguistic therapy, including but not limited to semantic and phonological factors in picture-naming (*Nakagawa et al., 2019*; *Shrubsole et al., 2017*). One future direction is to find links between graph measures and behavioral performances (*Meier, Johnson & Kiran, 2018*; *Meier, Kapse & Kiran, 2016*; *Palva et al., 2010*). Another future work is to find associations between variables embedded in stimuli, such as semantic components based on semantic feature norms (*Feng, 2015*), and graph measures of hothubs and coldhubs. Then, the speech-language pathologist would be able to arrange brain modifying stimuli during therapy.

## CONCLUSIONS

In this study, we investigated the activation–connection correlations for the performance of naming tasks by persons having post-stroke aphasia compared with healthy subjects. Although this study observed a small sample of participants with regards to a simple naming paradigm, it expanded our knowledge of hothubs by integrating both the activation and connection. The results reveal that the examined patients utilized different activation–connection coupling patterns and different sets of hothubs as healthy participants to perform the same task, thereby functionally reorganizing their brains. The findings also indicate that there were hubs with low activations in functional brain networks. Thus, dark functional networks must be considered in functional brain-imaging studies. The operational concepts of ''hothubs'' and ''dark functional networks'' will promote further development of target-selection approaches to therapeutic brain stimulation.

## ACKNOWLEDGEMENTS

We appreciate Dr. Chun FENG from the University of Nebraska Medical Center, for her kind editorial assistance with our manuscript.

### Funding

This work was supported by the National Nature Science Foundation of China (81672255), Jiangsu Higher Institutions' Excellent Innovative Team for Philosophy and Social Sciences (2017STD006), the Priority Academic Program Development of Jiangsu Higher Education Institutions (JX10231801), and the Hospital Construction Fund on Key Clinical Specialty of the Affiliated Sir Run Run Hospital of Nanjing Medical University (YFZDXK02-7). The funders had no role in study design, data collection and analysis, decision to publish, or preparation of the manuscript.

### Grant Disclosures

The following grant information was disclosed by the authors:
National Nature Science Foundation of China: 81672255.
Jiangsu Higher Institutions' Excellent Innovative Team for Philosophy and Social Sciences: 2017STD006.
Priority Academic Program Development of Jiangsu Higher Education Institutions: JX10231801.
Hospital Construction Fund on Key Clinical Specialty of the Affiliated Sir Run Run Hospital of Nanjing Medical University: YFZDXK02-7.

### Competing Interests

The authors declare there are no competing interests.

### Author Contributions

- Feng Lin conceived and designed the experiments, performed the experiments, analyzed the data, prepared figures and/or tables, authored or reviewed drafts of the paper, and approved the final draft.
- Shao-Qiang Cheng, Dong-Qing Qi, Qian-Qian Lyu and Li-Juan Zhong performed the experiments, authored or reviewed drafts of the paper, and approved the final draft.
- Yu-Er Jiang performed the experiments, prepared figures and/or tables, authored or reviewed drafts of the paper, and approved the final draft.
- Zhong-Li Jiang conceived and designed the experiments, authored or reviewed drafts of the paper, and approved the final draft.

### Human Ethics

The following information was supplied relating to ethical approvals (i.e., approving body and any reference numbers):

This study was approved by the Ethics Committee of The First Affiliated Hospital of Nanjing Medical University (2011-SRFA-025).

### Data Availability

Data are available at Zenodo: LIN Feng, & JIANG Zhong-Li. (2020). Functional Brain Networks of Picture Naming in Broca's Aphasia and Healthy Controls (Version v.20200114) [Data set]. Zenodo. http://doi.org/10.5281/zenodo.3831116.

## Supplemental Information

Supplemental information for this article can be found online at http://dx.doi.org/10.7717/peerj.10057#supplemental-information.

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
