# Peer review of "Brain hothubs and dark functional networks: correlation analysis between amplitude and connectivity for Broca’s aphasia"

_PeerJ, doi:10.7717/peerj.10057_

## Round 0.1 · original submission · Major Revisions

The reviewers have raised some very important concerns about your manuscript, which call into question the adequacy of your methodological and statistical descriptions. The manuscript would only become suitable for publication if this missing information can be provided in full. In particular, you need to ensure that all your claims are based on proper statistical comparisons (for example, the comment about permutation testing from Reviewer 2), that all figures are properly labelled and explained, that your methods are described in enough detail to allow replication and that all your design decisions are explained and justified. While these concerns would often be a reason to reject a paper, the reviewers also see strengths in the study. I, therefore, wanted to give you the opportunity to re-submit; but please note, it will only be worthwhile doing so if you feel this missing information can be added in its entirety, and if there are still interesting findings that emerge after proper statistical testing is used to assess your claims.

Reviewer 1 ·

Basic reporting

The paper is generally well-written and structured. The authors collected valuable MEG data in both patients and healthy control groups, although the sample is limited but the population is well-matched. The current study is focusing on the relationship between local activation and connectivity/network measures which is an important question and will be of wide interest to many researchers in the field of cognitive neuroscience.

Experimental design

Please describe the experiment paradigm with more details to help readers to fully understand how the picture naming task was conducted in the MEG scanner, for example, the authors mentioned that there were 40 stimuli in the experiment, and each run has 100 trials, how many runs (trials in total) do you collect?

Validity of the findings

1). Please provide more information on the behavioral performance in the MEG scanner, were there incorrect trials for the task in two groups, and if this is the case, was there a significant difference in accuracy between groups? And, did the analysis include both correct and incorrect trials? These behavioral effects could also be the reasons caused the difference of amplitude-connectivity relationships between groups.

2). One of the participants whose age at onset was 19 (B5), who was significantly younger than other participants in both groups, as extensive prior fMRI studies revealed that older adults consistently show decreased activity in visual cortex and medial temporal lobes in a diverse of tasks such as attention, visual perception, working memory(Cabeza et al., 2004; Gutchess et al., 2005; Maillet & Rajah, 2014; Pinal, Zurron, & Diaz, 2015; Rajah & D'Esposito, 2005; Spreng, Wojtowicz, & Grady, 2010). Also, other studies suggested there was over-activation in the prefrontal cortex in aging (Maillet & Rajah, 2013; Spreng et al., 2010). This could be a confounding factor of results reported here, especially considering that the limited sample in this dataset. Therefore, could the authors explain the reason why they included this young participant? Could the authors provide data to confirm the results were not driven by this participant?

3). In the abstract, the authors mentioned ‘Partial correlation’, I might be lost and not clear on which factor the authors want to control in the analysis. Could the authors provide this information in the Methods part?

4). Given the Broca group damaged parts of their brain, could this cause the amplitude differences in those areas and their connectivity patterns with other regions significantly different from control group? In other words, would the amplitude-connectivity relationships between groups be robust and stable after removing damaged areas in both groups, as those values, to some extent, might be outliers (in Broca group) that could cause the differences between two groups. This might be a wrong question since I am not familiar with lesion studies.

Cabeza, R., Daselaar, S. M., Dolcos, F., Prince, S. E., Budde, M., & Nyberg, L. (2004). Task-independent and task-specific age effects on brain activity during working memory, visual attention and episodic retrieval. Cerebral Cortex, 14(4), 364-375.
Gutchess, A. H., Welsh, R. C., Hedden, T., Bangert, A., Minear, M., Liu, L. L., & Park, D. C. (2005). Aging and the neural correlates of successful picture encoding: frontal activations compensate for decreased medial-temporal activity. Journal of cognitive neuroscience, 17(1), 84-96.
Maillet, D., & Rajah, M. N. (2013). Association between prefrontal activity and volume change in prefrontal and medial temporal lobes in aging and dementia: a review. Ageing research reviews, 12(2), 479-489.
Maillet, D., & Rajah, M. N. (2014). Age-related differences in brain activity in the subsequent memory paradigm: a meta-analysis. Neuroscience & Biobehavioral Reviews, 45, 246-257.
Pinal, D., Zurron, M., & Diaz, F. (2015). An event related potentials study of the effects of age, load and maintenance duration on working memory recognition. PLoS One, 10(11), e0143117.
Rajah, M. N., & D'Esposito, M. (2005). Region-specific changes in prefrontal function with age: a review of PET and fMRI studies on working and episodic memory. Brain, 128(9), 1964-1983.
Spreng, R. N., Wojtowicz, M., & Grady, C. L. (2010). Reliable differences in brain activity between young and old adults: a quantitative meta-analysis across multiple cognitive domains. Neuroscience & Biobehavioral Reviews, 34(8), 1178-1194.

Reviewer 2 ·

Basic reporting

No captions for supplementary figures or tables.
References appeared in the supplementary table 2.
No figures for paradigm or behavioral results.
I can't find any information about 'Raw data shared' in the main text.

Experimental design

Methods are not very detained, and I don't think the information is sufficient enough for later replication.

Validity of the findings

The statistics in the study are not sound and well controlled.

Additional comments

The manuscript by Lin er al. reports an MEG study aimed at investigating the source amplitude-functional connectivity relationship and its difference between Broca's aphasia and healthy control during a picture-naming task. To this end, the authors did a correlation analysis between the source amplitude and network connectivity that is indicated by several measures from graph theory analysis. They found a decoupling between amplitude and network connectivity. And this relationship showed different patterns for Broca's aphasics and healthy control group.

I have a few major concerns as followings:
1. The most important result for this study was the comparison between aphasia and control group on the amplitude-connectivity relationship, which was not done by statistics but by 'visual inspections' (Line 294). In the 'statistical analysis' section (Line 263), there were statistics on 'inter-group for amplitude' and 'within-subject Pearson coefficients', but none on 'inter-group Pearson coefficients', which should be the key statistic for this study question. I suggest to do a permutation test on inter-group correlation coefficients.
2. No behavioral data was showed for both groups. Then how to make sure that the aphasics paid attention and were involved the task? If not, the cognitive processing was not comparable between aphasia and control group.
3. Why choose PLVs for gamma band as the edges in graph modeling? Studies have shown that alpha and beta activity were involved in picture-naming task, for instance "Piai, V., Roelofs, A., Rommers, J., Dahlslätt, K., & Maris, E. (2015). Withholding planned speech is reflected in synchronized beta-band oscillations. Frontiers in Human Neuroscience, 9, 549." Another concern is that high frequency activity like gamma hardly shows stable PLVs across long-range neural networks. It's better to first show a significant gamma PLVs during picture-naming compared with baseline to assure it as a meaningful input for graph modeling.
4. Did equal number of trials enter the PLVs analysis for both groups? Because different trial number results in different signal-to-noise level for PLVs. I can't find this information in the manuscript, if not, I suggest to redo PLVs with equal number of trials for both groups to eliminate the bias.
5. 'Dark functional networks' is a key concept in this manuscript, but no specific network names were given for it and no clear discussion on its cognitive meaning.
6. It'll be great to show some links between graph modeling measures and behavioral performance (e.g., RT). Please see this paper as a reference "Palva, J. Matias, et al. Neuronal synchrony reveals working memory networks and predicts individual memory capacity. Proceedings of the National Academy of Sciences 107.16 (2010): 7580-7585."
7. No enough methods details for later replications:
1> No illustrator for paradigm, where is the naming production period?
2> 40 pictures for 100 trials (Line 107), which pictures are repeated? Does this repetition affect naming performance? 100 pictures per run (Line 115), how many runs in total for each participant? What is the content of these pictures? It's better to show a few example pictures.
3> what's the zero point in the epochs (Line 148)?
4> what is the amplitude data, is it raw data or filtered data?
8. For figures and tables:
1> No captions for supplementary figures or tables.
2> References appeared in the supplementary table 2.
3> Subscriptions in table 2 were hard to be seen.

---

## Round 0.2 · Minor Revisions

I can see that both reviewers are now largely happy with the manuscript but the permutation testing approach needs to be described in more detail. When preparing this revision, please double-check that the manuscrpt is as easy to follow as possible, and that all relevant details of the analysis as well as the data collection are reported in sufficient detail to allow replication.

I should be able to assess these changes myself, without sending out for re-review.

Reviewer 1 ·

Basic reporting

no comment

Experimental design

no comment

Validity of the findings

no comment

Additional comments

The authors have fully addressed all my comments, I found their responses quite satisfactory
and the revised version has been much improved.

Reviewer 2 ·

Basic reporting

no comment

Experimental design

The revised version added a paradigm figure and more detailed descriptions of the experimental design, which makes the manuscript more clear to be read and easier to be replicated by other researchers later.

Validity of the findings

The authors added a permutation test for the inter-group contrast analysis that increased the statistical power of the study.
But could the authors elaborate the permutation test in more detail, like which variable was shuffled?

---

## Round 0.3 · accepted · Accept

Thanks for making these additional revisions to the manuscript. I am delighted to accept your interesting study for publication.